# Methodology to Investigate Indigenous Solid Waste Systems and Practices in the Rural Areas Surrounding Maseru (Kingdom of Lesotho)

**DOI:** 10.3390/ijerph18105355

**Published:** 2021-05-18

**Authors:** Mpinane Flory Senekane, Agnes Makhene, Suzan Oelofse

**Affiliations:** 1Environmental Health Department, Faculty of Health Sciences, University of Johannesburg, Johannesburg 2028, South Africa; 2Nursing Department, Faculty of Health Sciences, University of Johannesburg, Johannesburg 2028, South Africa; agnesm@uj.ac.za (A.M.); SOelofse@csir.co.za (S.O.); 3SMART Places Cluster, Council for Scientific and Industrial Research, Pretoria 2000, South Africa

**Keywords:** indigenous, practices, environmental impacts, health impacts, rural communities, solid waste management systems, waste

## Abstract

Solid waste management (SWM) is the greatest challenge facing environmental protection and human wellbeing in the rural communities of Maseru (Kingsom of Lesotho). A lack of formal waste management (WM) systems in rural areas of Maseru have resulted in different indigenous systems and practices of SWM. Direct observation and descriptive designs will be employed. This is a mixed methods study of qualitative, quantitative and, non-experimental. We obtained data sets from existing official census and statistics of Maseru. We sampled 693 participants from total population of 6917. We received ethical clearance from Research Ethics committee of Health Sciences at the University of Johannesburg, we recruited six field workers. We have preventive equipment (sanitizers, masks, and sterile latex gloves) for COVID-19 infections in place; we have specific design on caps, masks and bags that will identify field workers as they collect data. We will train field workers, administer questionnaires, interview, and observe participants. STATKON will analyse data. The research will share the results with the Ministry of Environment and the community in Lesotho. The results will also be used to educate the rural communities on improved WM. Where weaknesses are identified, mitigation measures can be evaluated and implemented to rectify the negative aspects and improve the systems and practices. The rural communities face challenges such as waste collection services and sanitation facilities and this fact points out that there is a gap in SWM, which favours the existence of indigenous systems and practice of SWM.

## 1. Introduction

Numerous factors contribute to practices of indigenous SWM. The absence of formal collection systems in the rural communities are such that it is impossible for rural households to manage generated waste in a manner that does not pose a threat to the environment and human wellbeing. Many rural areas still practice historical methods of managing waste such as burying or burning it. These practices release harmful pollutants, which have negative effects on human health and the environment. Uncontrolled waste disposal that contaminates water sources and land seem to be the most common practice in the rural areas. Lack of knowledge about health impacts on humans and the occurrence of infectious diseases are potentially associated with uncontrolled solid waste (SW) [1]. The rural communities face challenges such as lack of waste collection services and sanitation facilities. This fact points to a gap in SWM, which favours the existence of indigenous systems and practice of SWM [2]. Households in the rural areas generate hazardous waste such as chemical fertilisers, pesticides, cleaning agents used in animal husbandry and insecticides, which pose a risk to man and the environment if not managed or disposed of properly [2]. WM is considered as one of the basic human rights that is linked to the Sustainable Development Goals, and in line with this, there is growing pressure on both urban and rural areas in both developed and developing countries to improve SWM. People read books and articles to understand and draw conclusions on published work dealing with a specific research topic [3]. To this study, literature will be introduced with the aim of providing context and background to the study, which aims to provide critical analysis of indigenous systems of WM in the rural communities of Maseru in Lesotho. The literature reviewed includes that which deals with indigenous practices of WM in developing and developed countries. The central focus of this study is the analysis of indigenous systems of WM in twenty rural areas surrounding Maseru. It is important to understand the entire waste stream and their characteristics for monitoring and control purposes, thus, this section discusses the characteristics of waste generated in rural communities surrounding Maseru. Ref. [4] state that waste in the rural areas is characterized by the culture and practices of the specific communities. According to [5], the characteristics of waste are analysed by categorising the components into degradable and none-degradable wastes. Ref. [6] state that degradable waste includes waste generated in the kitchen and in the garden while non-degradable waste includes plastics, linen, paper, glass, batteries, electronic waste and, scrap metals. The waste is further split into physical and chemical characteristics.

### 1.1. Physical Characteristics of Solid Waste

It is imperative to understand the solid waste stream characteristics such as density, calorific value, moisture content and composition so that waste generators and waste handlers know how to manage the waste through suitable methods [7]. Understanding the density of different types of waste provides an estimation of the space required for disposal in the landfill site in areas where there are municipal SW services [8]. In areas where incinerators are used to treat solid and render it free from microorganisms, the design and the operation of the incinerator determines the heating value of the waste to be treated, it is important therefore to first consider the calorific value of the waste because the entire process of waste treatment is influenced by the physical characteristics of waste [9]. Moisture content in SW depends on the type of waste, climate conditions and the composition of the waste which differ by lifestyle of communities, social parameters, season and by the place of formation [7]. The waste composition differs with the economic development and the standard of living in each area. The literature has shown that low-income communities generate waste that is high in moisture content [10]. The high moisture content waste makes it difficult or impossible to employ appropriate technology of managing waste, thus in areas where waste generated contains a lot of moisture, there are numerous open dumping of SW which occurs because municipalities have no budget to reduce the moisture content of waste before disposal, thus, moist SW that is not controlled have had adverse effects on the environment and the public health [10]. Physical characteristics of SW include concentration of heavy metals (HM) such as copper (Cu), nickel (Ni), lead (Pb), zinc (Zn), chromium (Cr), iron (Fe), manganese (Mn) and cadmium (Cd) [11].

HM are a hazardous component of general waste and are found in rubber, paper, and cement. These metals are toxic even at low concentration where some of them like Cr can cause skin irritation and ulceration [12]. Since they are contained in general waste, HM are discharged with dust and indirectly via sewage sludge to the environment. Metals may also be collected by waste reclaimers for recycling, on different situation may be disposed of in the landfill site or transported to municipal incinerators where it is treated before disposal as ash. Small amounts of HM are disposed of as chemical waste [13]. According to [14], a number of applications that include technology, agriculture, industrial, domestic and medical have raised concern over potential effects on environment and humans because of their distribution to the environment. HM are considered systematic toxicants even at lower concentrations because of their contribution to inducing multiple organ damage [15] postulate that Cr is a HM used in preserving wood, glassware cleaning solutions, metal finishing and in leather industries. In line with this, many people that are found to be working in these occupational setting are exposed to Cr-borne fumes, salts, dust and mist [16] state that because a number of large industries are disposing of their untreated HM into fresh water, there is a great public concern as these HM are generally toxic to aerobic and anaerobic processes as well as animals.

### 1.2. Chemical Characteristics of Solid Waste

Typical chemical characteristics of SW include ignitability, corrosivity, reactivity and toxicity. Ignitable waste is categorised into four as (1) liquids that have a flash point below 60 °C, examples are alcohol solutions, HPLC liquids and glass cleaning solvents rinses (2) Spontaneously combustible solids such as metal powders and activated charcoal (3) Ignitable compressed gases; these are gas cylinders that are used in the laboratories (4) Oxidizers like peroxide compounds, perchlorate compounds and nitrate compounds [17]. Corrosive waste is a liquid form of waste that has a pH <2 or >12.5 and this includes (a) inorganic acids such as hydrochloric, nitrate, sulphuric and phosphoric (b) organic acids such as acetic and lactic (c) Alkaline compounds such as amines and hydroxide [18]. Reactive waste is categorized into five compounds of (a) compounds that violently react with water (b) compounds that are capable of detonation or explosive reaction (c) compounds which are unstable and readily undergo violent change without detonating (d) Cyanide and sulphide-bearing compounds (e) Compounds which form potentially explosive mixtures with water-metal hydrides and calcium carbide. Toxic wastes which are categorized into three as heavy metals (mercury, lead and barium), pesticides (toxaphene, chlordane, lindane) common organic chemicals (benzene, methyl ethyl ketone and chloroform). References [19,20] indicate that Gomez et al. analysed household SW for the chemical characteristics and found that SW contains moisture, ash, heavy metals, volatile solid and calorific value. The chemical characteristics of SW are important to assess possible options such as energy content and fusing point of ash in waste processing and waste recovery.

## 2. Literature Review

WM remains a challenge in the Kingdom of Lesotho. The types of wastes known to be generated are mainly from textile industries, which remain the largest for their immense contribution to the economy of Lesotho, brewery and soft-drink industry, commercial industry, tannery industry and residential households [21]. Waste generated in Maseru in 2012 was mostly recyclables, the types included plastic (63.50 tons), can (67.24 tons) paper (261.35 tons) glass (691.77 tons) and cardboard (370.45 tons). Scrap metals are recyclables, and they have significant monetary value. Scrap collectors such as Lesotho Scrap Metal Dealer collected generated scraps to a total of 4469.23 tons and Welcome Transport collected generated scraps to a total of 262.23 tons. The study that was conducted by [6] shows that high (55–80%) quantities of waste are generated from households and 10–30% of waste is generated from commercial sources [4] state that in most developing countries such as Lesotho, newspapers and other clean papers are used for packaging fruits and vegetables. Other sources include old cars, scrap metal, litter, and effluent from textile manufacturing industries in Maseru urban areas. Maseru is facing SW crises due to large quantities of waste generated daily, and there is an urgent need for SW specialists and statisticians to assist policy makers in Lesotho to develop policies and implementation programs [22]. In Lesotho, SW is insufficiently managed, and this is an enormous public health concern because some of the waste has been proved highly infectious and toxic if children are exposed to it. In 2015 at Qoaling Township in Maseru, a toy injured children who were playing with dumped waste when it exploded. Households waste is observed scattered within Qoaling Township, footpaths and along the streets. In another incident at Lithabaneng Township in Maseru, a medical waste was found mixed with commercial and industrial waste [23]. “Poor SWM contributes to increased concentrations of heavy metals in aquatic ecosystem, raising concerns because of its toxicological impacts on aquatic ecosystem and associated human health risk [24]”. Ref. [24] show that poor SWM generated from industries and households poses a pollution threat to the Maqalika Reservoir in Maseru. This reservoir serves the Maseru inhabitants with drinking water and fish. Heavy metal sedimentation and concentration levels account for the primary exposure of human health [18,24] states that land development practices in Lesotho should be incorporated into indigenous practices of SWM by creating a tag called “Thotobolo” which means “Ash heap”. Thotobolo is a SW disposal area in each household yard that should not be considered or classified as landfill site because of its location, size and how it is managed by each individual rural community. The land planners should know where it is within each household yard and understand the causes, which then should be aligned with the present-day land development practices. The reason is that Thotobolo is used for many traditional practices such as applying ash on human anatomical organs like legs and feet to traditionally heal unknown conditions and diseases. Besides, the ash is also applied on windows to symbolize death in the family.

The study that was conducted by [6] shows that high quantities of waste (55–80%) are generated from households and 10–30% of waste is generated from commercial sources. According to [25] Ghana generates 0.09 kg/day while Seychelles generates more waste than Ghana to a volume of 2.98 kg/day per household. South Africa, Egypt, and Nigeria, however, have become the highest waste generating countries in African continent with volumes ranging from 23 to 21; 18 to 35 and 17 to 45 million tons per year. “Indigenous systems of WM refer to local practices that are unique to a given culture or society. It is a traditional science and traditional wisdom that is confined to a particular culture or society [26]. Indigenous systems and practices of SWM becomes a challenge to some rural communities when they are confronted with environmental pollution. The spread of pollution affects the lives of everyone and the culture of the community because it challenges their ability to sustain their identity, rights and responsibility in conserving their environment [26]. African rural communities have diverse culture. Lesotho has a diverse culture but in the rural areas people live a traditional way of life. The understanding of this is rooted from their stories, myths, beliefs, taboos, and proverbs. According to [26], rural communities that live in proximity with natural resources tend to encompass WM systems that govern natural resource use. Such communities believe that their natural environment can be conserved successfully through utilising their tradition, cultural mores, and practices. The negative impacts of indigenous system of WM include diseases that are caused by various vectors such as rodents and mosquitoes. In addition, offensive odours from heaps of waste that is improperly managed affect human beings. These problems are on one hand the origins of ignorant waste generators and on the other hand, lack of improved drinking water sources, sanitation, and absence of WM services [27]. African rural communities have diverse culture. Lesotho has a diverse culture but in the rural areas people live a traditional way of life. The understanding of this is rooted from their stories, myths, beliefs, taboos, and proverbs. According to [27], rural communities that live in proximity with natural resources tend to encompass WM systems that govern natural resource use. Such communities believe that their natural environment can be conserved successfully through utilising their tradition, cultural mores, and practices. Culture in some rural communities in developing countries is based on verbal communication and practical application through activities. Participants in these activities listen to authorities and watch those demonstrating activities then act. This is because of high illiteracy rate in some developing countries [28]. According to [29] growth in economy and urban populated are both linked to how municipal SW is managed. On a global scale, the composition of waste generation in different countries depends on high, middle and, low income levels. The imbalance of income among different groups has influence researchers and expects in SW to identify appropriate SW treatment methods to solve the gap between income and operating costs, thus in low income communities, the best option to manage waste effectively is to avoid utilizing large quantities and expensive resources but to separate it at the point of generation and to recycle it to avoid waste that goes to the landfill site [30] concur with [29] and shows that while management of waste is a complex task, decision makers should protect the environment from adverse effects caused by poor SWM by employing technology to safe costs and improve waste recovery [31] state that municipalities should supply communities refuse bins. Recycling stations should also be made available for every member of the community to prevent illegal dumpsites. Availability of recycling stations will create jobs for unemployed and will contribute to cleaner environment because no waste will be disposed of on the street, hence, a win towards sustainable development and green world [32] postulate that there is a need for waste-to-energy facilities that should be in suitable areas closer to the communities, facilities that will not only minimize transportation costs but also allow for large investments and long-term projects. To achieve these, transportation and revenue models should be developed [32] further point out that economic performance may be met if the facilities are fully utilized by generating a revenue through standard gate fees for facilities users.

### 2.1. The Impact of Indigenous Systems of Solid Waste Management on the Environment

Most waste in the rural areas is poorly managed and this affects the environment in which we live negatively. Poor management of waste is due to rapidly increasing population and it is aggravated by non-existence of traditional management systems of waste [33]. The negative impacts shown in Figure 1 include release of methane gas, atmospheric pollution, the contamination of surface and ground water. Therefore, there is a need to address an understanding of urban and rural communities’ behaviours through concerted efforts to find ways to improve indigenous WM. “Comprehensive studies on public behaviours and waste generation in urban-rural areas are among few fields that demand endless emphasis if a sound management was to be reached [34]”. In the past, the rural communities had only biodegradable SW that was traditionally well managed by employing techniques such as composting and manuring. However, the amount of generated SW from various sources such as manufactured goods has increased and is creating problems on the environment in which rural communities live. Unlike in the urban areas, the rural communities are not able to manage SW in an appropriate manner because of scarce experts, lack of financial support and lack of technological capacity [28].

There are several international agreements towards addressing environmental pollution at a global level. For instance, the theme “Towards a pollution-free planet” was aimed at controlling and abating environmental pollution. “The Aichi Target 20 of the Convention on Biological Diversity’s strategic Plan for Biodiversity 2011–2020 was to reduce pollution to levels that are not harmful to functioning of ecosystem and biodiversity by 2020, while the sustainable development goals emphasised an urgent need to address environmental pollution in several of its targets”. Considering these, environmental impacts seem to be unequally distributed with rural communities being most affected [35]. In a study that was conducted by [36] in Northern Canada on the impacts of indigenous SWM systems on the environment, it was found that natural projects shaped the infrastructures and lifestyles of the rural communities of Asubpeeschoseewagon Netum Anishnabek. The waste disposal system employed was identified as a major threat to the environment because chemical waste such as mercury was disposed of directly into English-Wabigoon river. Major concern of environmental thread was expressed by Labrador rural community in North Canada who complained about a similar problem because their self-identities are formed by the land and natural attributes. Mercury contaminated main water sources, this led to municipalities in these areas used municipal trucks to provide the communities with drinking water, this was often disrupted by unsatisfactory weather conditions. The situation forced the healthy waters Labrador to promulgate critical waste issues which included lack of sewage and wastewater purification facilities and the need to improve management of indigenous SW [37] in a study conducted in African countries such as Kenya, shows that “Decomposition of SW in open spaces, uncontrolled dumpsites or storm water drainage and open burning of waste are likely to negatively impact the environment, including the pollution of soil, water (fresh and marine) and air. Some waste also contains toxic chemicals (heavy metals) and persistent organic pollutants (POPs), which are persistent in the environment, can travel long distances, and are likely to accumulate in fauna and flora and in the food chain”

### 2.2. The Impact of Indigenous Systems of Solid Waste Management on Human Wellbeing

Figure 2 shows the negative impacts of indigenous systems of WM which include diseases that are caused by various vectors such as rodents and mosquitoes. In addition, offensive odours from heaps of waste that is improperly managed affect human beings [2] There is evidence of a link of indigenous systems and practices of SWM with the living environment and health of the rural communities and this evidence shows that poor environmental health is related to exposure to biomass smoke and water that is contaminated by heavy metals. The impact of SWM on health of communities in the rural areas has received a great deal in the academic research because it is not well understood, and it is often perceived not as a public health problem but an aesthetic problem. Potential health issues may occur because one was exposed to environmental hazards found in waste the association of which is with every phase of WM from the point of generation to final disposal [38].

The rural communities experience poorer health than the urban communities. This is an inequality that has been observed in many countries [39]. Despite the millennium Development Goals, the world in which we live is faced with challenges that are caused by human activities. The activities include factors such as indiscriminate disposal of SW, lack of SW segregation at the point of generation, and open burning of SW. These factors affect the environment negatively where land and air pollution feature as main risks to human health. Other human health risks factors include contaminated water sources, poor sanitation and a burden of disease. In light of these, we conclude that there are people who are susceptible than others [39]”. In a study that was conducted by [40], it was found that indigenous SW has a negative impact on the health of the rural communities. The spread of disease resulted from blocked or constricted waterways by mosquito breeding in stagnant water and waste canals; SW workers and residents near illegal dump sites were at risk of contracting diseases due to the existence of flies, insects, and vermin. Furthermore, SW workers were exposed to odour and flies and contaminated potable water due to polluted rivers with solid waste. Ref. [37] states that in Africa, the impacts of SW are weak. In many rural communities, there are no municipal waste collection services and waste is found on streets, near houses and in drainage systems. The impacts of uncontrolled SW depend on the type of generated waste, how waste is disposed of and the duration of exposure to human beings. People who are exposed to may suffer sever morbidity and disability while other may even die. Electronic waste can be dangerous to children who trawl through dumpsite because it contains hazardous chemicals such as mercury and lead. Health impacts include those that affect the respiratory system if human beings inhale toxic substances. Mosquitoes found in waste carry organisms that cause malaria which in turn affects human wellbeing negatively [41] conducted a study in Ghana and found that many rural communities such as Sawaba were at the time of the study faced with SWM and health problems that undermine efforts to ensure good health and safe environment for all. Communities who were living next to open dump spaces had contracted diseases such as malaria and skin infections because of indiscriminate disposal of SW. Rotten organic waste became a breeding place for mosquitoes, heaps of waste were disposed of in unauthorised places, next to building sites, next to water channels. Communities were concerned because of mosquitoes breeding places because of indigenous systems of waste disposal.

## 3. Research Paradigm

Ref. [42] Define research paradigm as a guideline used to construct and implement research systems. According to [43], research paradigm uses ontology, epistemology, and methodology to define the nature of a phenomenon. An indigenous research paradigm believes that knowledge is relational, therefore; the researcher should consider sharing knowledge that will emanate from the finding of the study with participants, government of Lesotho and the University of Johannesburg. The Philosophy that underpins this paradigm is the Critical Theory and informed knowledge on indigenous systems [44].

### Indigenous Research Paradigm

The indigenous research paradigm (IRP) is a shared worldview that focuses on beliefs and values of social groups that are historically oppressed. The shared world serves as a guideline for solving the problems of the social groups [44]. The concept of indigenous is described as a perception of a place-based human ethnic culture that has not migrated from its motherland and is not a habitat or migrant population [45]. Literature has shown that the concept of indigenous is interpreted in different ways by different scholars, thus, this study uses the concept of indigenous to mean “Traditional way of doing things” and links this concept with indigenous research paradigm. The relationships between the people, past, present, future, living and none-living things is recognized in indigenous research paradigm that emanated from indigenous value systems [46]. For this study, the notion of relationships forms a basis for a researcher to interrogate the patterning of social life of the rural communities surrounding Maseru and their connectivity to indigenous system and practices of SWM. The interrogation will serve to consider and engage a research paradigm within the local tradition that offers an approach to understanding the world. The researcher uses this paradigm discourse because the study area is situated at the interface of Western and indigenous systems and practices SWM. Literature has shown that communities living in these areas are indigenous, they believe in their culture, norms, standards, they value their land and resources. This study is guided by assumptions, believes, norms and values of indigenous research paradigm and it is important for the researcher to apply a relevant theory to know what each of these concepts mean. Theory that underpins indigenous research paradigm is a critical theory. The link between the research paradigm and the research philosophy is that the researcher assumes that the rural communities surrounding Maseru live a traditional style of life and they want to live that way. The communities believe that their indigenous environment can be conserved successfully through utilizing their tradition, culture, norms, values, beliefs all of which are contained in critical theory. This study assumes that the research questions and the community’s way of living in real life will assist the researcher to critically analyse and interpret the indigenous systems and practices of SWM in the rural communities surrounding Maseru by using the indigenous research paradigm and critical theory.

## 4. Research Design and Method

A research method is a strategy used to implement that plan [47]. This study will utilize a quantitative descriptive design and a direct observation research design method.

### 4.1. Research Design

#### 4.1.1. Descriptive Design

A descriptive research refers to the type of design that will be applied to a given topic [48]. To answer the research questions about the real–life situation in the rural communities of Maseru, the researcher will use a quantitative description technique [49]. Participants will also be interviewed and observed in their daily activities. Their responses will be assessed and critically analysed to understand their indigenous practices and systems of SWM. This method was chosen to investigate the background of a research problem and to collect quantifiable data necessary for statistical analysis of the sample used for the study.

#### 4.1.2. Direct Observation Design

Direct observation refers to a research design where the researcher collects data by observation what the participants do without altering that environment [50]. It takes a qualitative approach to observing the community individually or in groups while in their natural environment [51]. The researchers will comply with COVID-19 regulations by maintaining a social distancing of 1.5 metres while they observe participants in their daily SWM activities and document what is seen and heard in the field by using instruments such as the note pad, camera, and voice recorder. Camera and recording will be used with the permission of the participant. This design will assist the researcher to bridge the gaps in understanding the research problem; to interconnect with participants in their natural inhabited space; to gather additional data that support literature of the topic; to ask questions relevant to the specific cultural context and to realize new perspectives that take exception to the existing theoretical framework. The theoretical framework of this study derives from people’s indigenous systems and practices of SWM in the rural communities surrounding Maseru, Lesotho. Four theories that support this study include Critical theory, Health belief theory, theory of ecology and theory of planned behaviour. A critical theory and specific theories to key concepts for this study are addressed to explain the phenomena in the rural communities of Maseru and to provide possible solutions for how to respond to them. A critical theory is the work of collective Sociologists at the University of Frankfurt in Germany. This theory emerges out of the Marxist tradition of focussing on the relationships between economy, social structure, and social life [52].

### 4.2. Research Method

The method that will be used for this study assumes sampling from various rural communities of Maseru. This method involves questionnaires, interviews and, observations. Participants will receive questionnaires to answer quantitative and qualitative questions. Questionnaires are suitable to cover the research in twenty rural communities surrounding Maseru. To obtain detailed information about personal feelings, perceptions and opinions of participants, the research team will observe and ask participants questions based on what they will be doing at that time as they perform their WM activities. This will assist researchers to receive first-hand data. To gain cooperation of the participants, the researchers will engage in overt observation to inform the participants that they will be observed when performing their various WM activities. The researchers will stay away from the activities but will observe to understand the indigenous systems and practices of SWM in the rural communities of Maseru, Lesotho. The method that will be used for this study assumes sampling from various rural communities of Maseru. This method involves questionnaires, interviews, and observations. Questionnaires are suitable to cover the research in various constituencies of rural communities of Maseru. To obtain detailed information about personal feelings, perceptions and opinions of participants, the research team will conduct interviews. The researcher will receive first-hand data through observation. For this study, the researcher will engage in overt observation to inform the participants that they will be observed when performing their various WM activities. The researcher will stay away from the activities but will observe to understand the indigenous systems and practices of SWM in the rural communities of Maseru, Lesotho.

### 4.3. Study Setting

The setting for this study is within the three constituencies (Matsieng, Koro-Koro and Rothe) that together form the rural communities of Maseru district. From Matsieng and Koro-Koro constituencies, seven rural communities will be drawn randomly to participate in this study. Six rural villages will be drawn from Rothe constituency because the researcher was not able to find statistics for other villages therefore, the total number of village/rural community participants will be twenty. A pilot study will be carried out at Ha-Teko that is also a rural village in Maseru. The maps below in Figure 3, Figure 4 and Figure 5 show few of the rural areas that were selected for the study. A map that shows all areas under study was not found.

### 4.4. Study Population

Maseru is the capital city of Lesotho. The general population before January 2020 was estimated to 118,338 [50]. According to [53,54,55,56] the population estimates for villages under analysis were as follows: Matsieng constituency (Ha Leutsoa-39; Ha Moruthoane-31; Kholokoe-74; Ha Ramabele-21; Ha Rantsilonyane-28; Ha Mphafi-10 and Aupolasi-16). Koro-Koro constituency (Ha Makhalanyane-36; Phuleng Ha Makhalanyane-21; Ha Maja-57; Ha Sekete-26; Ha Mofoka-35; Molumong Ha Mofoka-10 and Aupolasi Ha Mofoka-39). Rothe constituency (Ha Mokaoli-57; Ha Thlakanelo-37; Ha Rasekoai-39; Mahuu-27; Leralleng-35 and Masite Nek-55). The total number of estimated study population is 6917.

### 4.5. Sample Size

Altogether, 693 respondents from the three constituencies in Maseru will be randomly selected from a total population of 6917 obtained from Statistics Lesotho office. An equal portion of 10% of the population in each rural community will be considered to represent the total population and to maintain homogeneity and neutrality in the analysis and interpretation of the study. The rural communities are under the leadership of a headman and he will assist the researcher to trace the respondents. Table 1 shows a list of rural communities identified to participate in the study, their estimated population and sample size for each rural community.

### 4.6. Sampling Method

Simple random sampling is a method that is used to allow every member of the population stand a chance to be part of the study [57] to conduct this method, the researcher will use random number generator to select the population. Families will be selected from rural villages that fall under three constituencies in Maseru district.

### 4.7. Recruitment of Participants

Rural communities are accountable to the village headman. The researcher will first have a meeting with the headman before commencing with data collection. The agenda points for the meeting will include the intensions of the researcher about the study in question; when will the study start? Who are data collectors, how will they be identified and how many are they? For how many days/months will the study be conducted? At what time of the day will the study be conducted and for how many hours? Who within the community members are expected to participate? The researcher will request the headman to randomly identify possible participants and then the researcher and the fieldworkers will go door to door to see if people are willing to participate. The door-to-door exercise will assist the researcher and the fieldworkers to identify the head of each household or the person who in-charge of cleaning and WM of the household. Domestic workers who are women assisting with house work and men who are working in the garden both generate, handle and manage SW, therefore are relevant people to be requested to be included in the study if they agree to participate.

### 4.8. Inclusion Criteria

This study will include adult males and females above 21 years of age. The ages are chosen because anyone within these ages is an adult and can be able to respond to questions without any problems. Participants will be selected randomly. The study will also include officials from the Ministry of Environment in Lesotho and the researcher will meet with them through focus groups to get their views on the situation. The researcher will send an email correspondence to request the meeting with the officials. The meeting will be held in the primary school immediately after data collection is complete in all identified areas. The researcher will play a role of a moderator to guide the focus group discussions; and will focus on probing into responses with the aim of finding out their motivations. In addition, the face to face discussion will uncover the real information that is hidden. One of the field research workers will assist the moderator with recording the discussion processes such as capturing key issues on a flip chart organised by the researcher. The researcher will ask open-ended questions to give responds free responses. Data collected from focus groups will be coded and analysed qualitatively and quantitively.

### 4.9. Exclusion Criteria

Community members (CMs) under the age of 21 years and over the age of 65 years, CMs who are blind are not able to use their vision in any meaningful way, visual impairment affects their ability to read printed material, thus reducing their ability to gain information and thereby making him more dependent on others. Deaf people may not know sign language and dumb will also not be included because they may not communicate effectively because of the defects of their vocal organs, and because of this, they may not be able to ask questions in the questionnaire where they need clarification. Visitors of less than one calendar month in the areas will be excluded because the researcher’s opinion is that they may not be able to respond to some questions.

### 4.10. Pilot Study

Fifty (*n* = 50) participants of Ha Teko in Maseru will be requested to participate in the study and they will not form part of the main study. Six fieldworkers will be trained before data collection. The researcher will conduct training for the fieldworkers on various issues that are relevant to the study for two days before the pilot study commences. Participants will be requested to respond to the same questions that will be administered to participants who will participate in the main study. This will assist the researcher to identify any problems with the questions and modify them before the main data collection commences. The pilot study will take two days because fieldworkers will work from 08:00 to 16:00 every day. Fieldworkers will spend 45 min with each participant to interview and observe them executing their daily activities that are related to indigenous systems of WM.

### 4.11. Recruitment of Fieldworkers

The study will be conducted in twenty rural villages and the researcher will need people who will be assisting with the administration of questionnaires to these communities, therefore, the researcher will recruit six fieldworkers with the assistance of The Public Service Placement (PSP) of Lesotho. PSP is a government section that is responsible for recruitment and employment of University graduates at all diploma and degree levels. The researcher will request PSP to assist with graduates in the disciplines of Environmental health, Environmental management, Education, Land planning and Nursing who do not yet have employment and are in their list of job seekers. The researcher will train fieldworkers for two days before they start with data collection. Arrangements will be made by the researcher to use a classroom in one of the primary schools in Maseru. A primary school is chosen because many of them close at 14hrs and after this time, a classroom can be utilised for training.

## 5. Data Collection

To understand the indigenous systems and practices in the rural areas of Maseru, Table 2 shows a self-developed questionnaire written in English that utilizes a descriptive research design will be utilized and it will be translated into Sesotho. The researcher and six fieldworkers will each do 116 interviews and will conduct six to seven interviews per day over 40 days. The households will be divided into groups and given a date in which they will be requested to participate in the study. This will make it easy for the researchers not to visit or repeat households that have already participated. The researcher will ask for permission to record the conversation on the researcher’s recording device before starting with interviews. Participants will sign a consent form if they agree. The researcher will go ahead with the interview without recording it if participants decline the recording but will keep as careful notes as possible. A total of 17 to 19 out of 693 questionnaires will be administered every day on Monday to Friday from 08:00 to 16:00. The timeframe for this is anticipated to be 40 working days and it will take place in April–May 2021. A pilot study will take five working days and it will start in March 2021 (two days is scheduled for training field workers).

### Example of Qualitative Questions

(1)In your opinion, what do the people in this community understand by indigenous systems and practices of SWM/Kutloisiso ea hao ke efe mabapi le kutloisiso ea baahi ba moo ea taolo ea lithole?(2)Explain your understanding of the impact that recognition of indigenous systems and practices of SWM in this community has in addressing local environmental issues/Hlalosa hore na kutloisiso ea hao ke efe mabapi le tsebeliso ea sekhale ea ho laola lithole e le ho araba litaba tsa tikoloho.(3)Describe the indigenous practice that the people in this community do to enhance communal WM/Hlalosa mokhoa oa khale oo baahi ba mona ba o sebelisang ele ho ntlafatsa taolo ea litsila ka kopanelo.(4)Describe the waste disposal systems that the people in this community use in order to prevent the breeding of mosquitoes and other health problems/Hlalosa mokhoa oa ho lahla litsila oo baahi ba mona ba o sebelisang ele ho thibela ho ikatisa hoa menoang le mathata a mang a bophelo ba baahi.

## 6. Data Analysis

According to [58], data analysis refers to a study of material data which has been organized to discover the underlying facts. Data will be analysed qualitatively and quantitatively. For quantitative analysis, data will be compiled to transform raw data into significant and understandable approach using the following three steps: data validation; data editing, and data coding. Data will be analysed in terms of findings on waste characteristics, indigenous systems and practices of SWM, the impact of SW on human wellbeing and the environment. Literature will form a base for discussion on the analysed data. For qualitative analysis, interview and observation notes will be coded and analysed separately because they are different kind of data. The researcher will create a separate document group and create a household number variable such as “Indigenous # 00” and “indigenous # 01” for all groups of documents. This will assist the researcher to recall and make a comparison of interviews and observation about the same household in the future by activating the household number variable. For these reasons, the researcher will seek assistance from STATKON (University of Johannesburg statistician) to analyse data. STATKON will use International Business Management Statistical Package for Social Science Version 25.0 (Johannesburg, South Africa). Quantitative method will be employed to interpret the results. Data management will be sorted by subscribing to Figshare available in the library of the University of Johannesburg to save raw data for five years, this will be done immediately when the study is completed.

## 7. Reliability and Validity

Reliability explains the extent to which a particular test, procedure, or data collection method (questionnaire) can produce similar results under same circumstances [59]. Closed-ended standard questions are set, they are categorised into twelve groups, and the rating is in a Likert scale format from 1 to 5. Each category will be analysed separately. Reliability of data will be analysed by using Cronbach’s alpha with computer software. Validity refers to whether research findings present a true and trusted view [60]. Validity is about determining whether the study is measuring what the researcher is intending to measure [61]. The researcher will develop standardized questionnaires that are based on findings of previous studies. Open and closed-ended questions will be set in such a manner that they will measure the indigenous systems and practices of SWM in the rural communities of Maseru. The sample size of 10% of the total population in each identified rural area will be used, this will be sourced from men and women of the ages between 21 and 65. This study will involve six research field workers and the researcher will ensure that field research workers phrase the questions the same way each time and this will be clarified during training of research field workers, the training will focus on to count a behaviour or responses. The researchers will ensure that there are no external influences that could possibly cause variation in the results.

## 8. Ethical Considerations

It is the researcher’s obligation to respect the participants therefore, the researchers use UJ research ethics guidelines and the following issues will be considered before, during and after the study. A letter of permission to conduct a study was obtained from Higher Degree and Research Ethics committees of the University of Johannesburg (NHREC registration: REC 241112-035). Prior to commencement of the study, participants will be informed of the purpose, nature, how data will be collected and the extent of the research. The questionnaire will have a tick box where a consent form can be indicated, a consent form will allow the researcher to involve them as participants in the study and will be provided with all the information throughout the study. They will be informed about the role of the researcher, and without being forced participants will give concerned to participate in the study. In this study, human beings are autonomous and have self-determination to make their own decision in participating in the study; therefore, data will not be obtained at the risk of harming participants. Participants will be treated with respect and dignity. In this study, no experiments or trials are involved, and no harm will occur to man, animals as well as the environment. Participants will be protected from harm and discomfort. They will be informed that there is no direct benefits for participants in participating in this study, which is indigenous systems and practices of SWM and the impact on the environment and human wellbeing. All participants meeting the selection criteria will be given a chance to participate in the study. They will be treated with fairness, dignity, and respect. The researcher will adhere to ethical standards to gain honesty and trust of the participants about data collected and analysed. The researcher will ensure that the questionnaire does not include any identifying characteristic such as participants’ names, contact details and physical address.

## 9. Possible Outcomes

The researcher will share the findings and recommendations with the Ministry of Environment in Lesotho as well as specific officials who were included in the focus group. The study will produce descriptive information about aspects of indigenous systems of managing waste that will be of interest to the community, stakeholders, and policy makers. The researcher will first share the results with the community and use them to educate them on improved WM. Presentations from the study will be made both nationally and internationally. It could be necessary for the researcher to review this study with the aim of assessing any developments in terms of shifting from indigenous to a modern system of managing waste in the rural communities of Maseru. Each research question assessed and critically analysed can help identify the specific weaknesses and strengths in indigenous systems and practices of SWM in the rural communities of Maseru. Where weaknesses are identified, mitigation measures can be evaluated and implemented to rectify the negative aspects and improve the systems and practices.

## 10. Conclusions

It is clear from literature review that there is a lack of formal WM systems and practices in rural areas of Maseru and this has resulted in different indigenous systems and practices of SWM emerging. These systems are currently not regulated and may have unintended consequences to the environment and human health. There is currently a general lack of knowledge on the practices and impacts (both positive and negative) of these systems on the environment and human wellbeing. In summary, indigenous systems and practices of SWM enables the indigenous community to collectively control their customary estates. Indigenous systems and practices of SWM supports culture, promotes norms and values of rural communities. It supports sustainable development through researchers. New technological methods for WM may be adopted to allow the rural community to utilize the waste items they generate to accommodate their needs. The indigenous communities understand ecology and sustainable development; however, they are resistant to participate in modern WM policies, thus creating a problem for local authority to concurrently implement the modern policies and consider the old age practices. Rural communities have their own way of living and doing things in favour of their culture, norms, and values, it may take time to make them change their style of living. Reality is, human health and the environment are affected negatively by indigenous systems and practices of SWM. The negative impacts on the environment include release of methane gas, atmospheric pollution, the contamination of surface and ground water. The negative impacts of indigenous systems of WM include diseases that are caused by various vectors such as rodents and mosquitoes. In addition, offensive odours from heaps of waste that is improperly managed affect human beings. There is evidence of a link of indigenous systems and practices of SWM with the living environment and health of the rural communities and this evidence shows that poor environmental health is related to exposure to biomass smoke and water that is contaminated by heavy metals.

## Figures and Tables

**Figure 1 ijerph-18-05355-f001:**
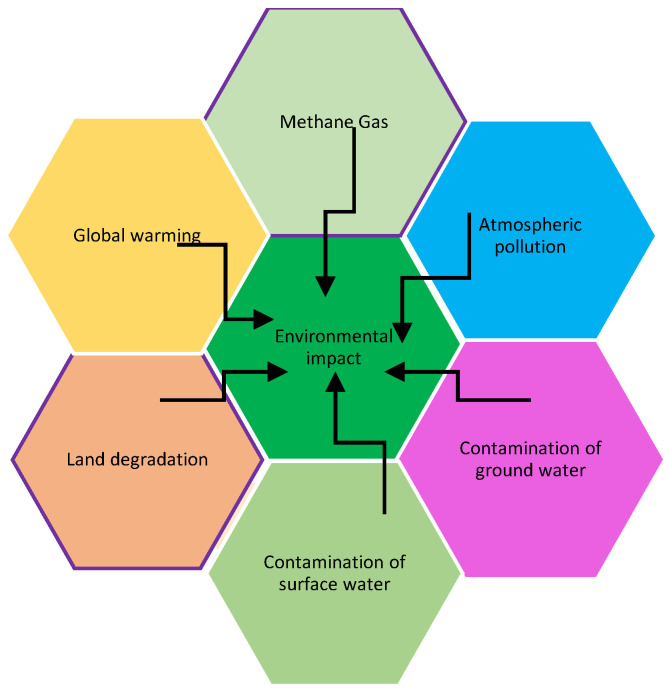
The impact of indigenous systems of solid waste management on the environment.

**Figure 2 ijerph-18-05355-f002:**
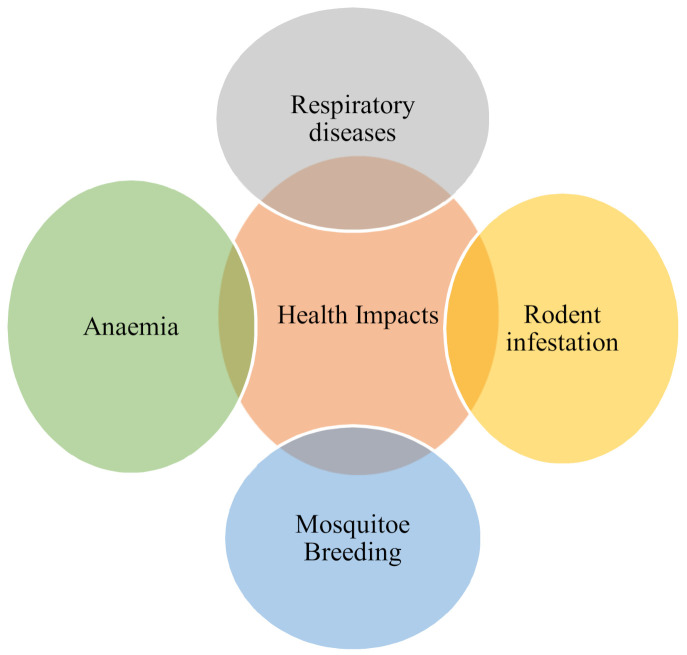
The impact of indigenous systems of solid waste management on human wellbeing.

**Figure 3 ijerph-18-05355-f003:**
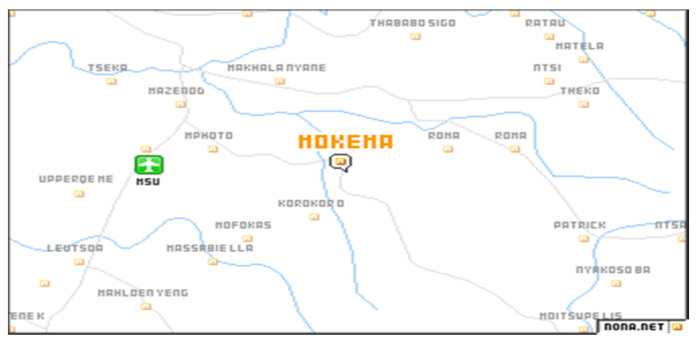
Map A shows Makhalanyane, Koro-koro, Mofoka, Leutsoa and Mahloenyeng. Source: [53].

**Figure 4 ijerph-18-05355-f004:**
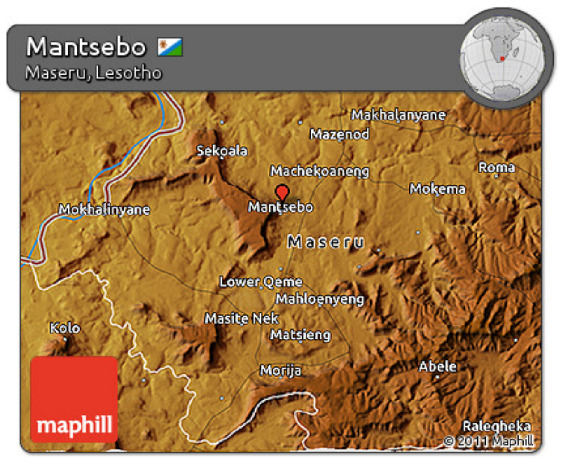
Map B shows Masite Nek, Mahloenyeng and Makhalanyane. Source: [54]

**Figure 5 ijerph-18-05355-f005:**
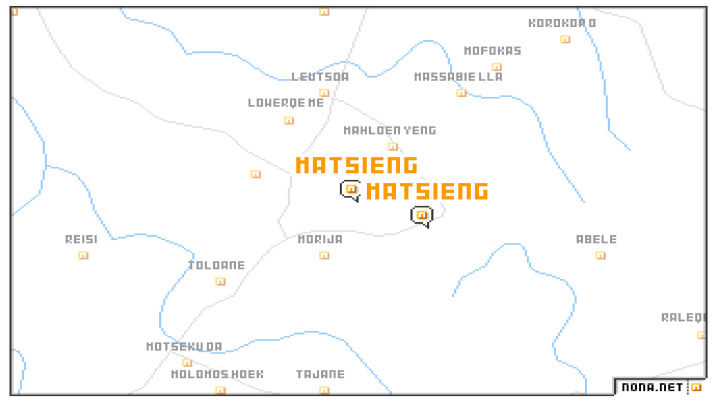
Map C shows Koro-koro, Mofokas, Leitsoa and Mahloenyeng. Source: [55].

**Table 1 ijerph-18-05355-t001:** List of rural communities of Maseru, their estimated population and sample size for the study [22].

Rural Community	Estimated Total Population	Sample Size
	Matsieng Constituency	
Ha Leutsoa	386	39
Ha Moruthoane	312	31
Kholokoe	735	74
Ha Ramabele	212	21
Ha Rantsilonyane	284	28
Ha Mphafi	98	10
Aupolasi Mahloenyeng	160	16
Sub-total	2187	219
	Koro-Koro Constituency	
Ha Makhalanyane	355	36
Phuleng Ha Makhalanyane	208	21
Ha Maja	569	57
Ha Sekete	260	26
Ha Mofoka	346	35
Molumong Ha Mofoka	98	10
Aupolasi Ha Mofoka	391	39
Sub-total	2227	224
	Rothe Constituency	
Ha Mokaoli	573	57
Ha Thlakanelo	365	37
Ha Rasekoai	394	39
Mahuu	270	27
Leralleng	35	35
Masite Nek	548	55
Sub-total	2503	250
Total	6917	693

**Table 2 ijerph-18-05355-t002:** Example of quantitative questions.

The Indigenous Systems and Practices Used in the Rural Communities of Maseru/Tsamaiso Le Mekhoa Ea Khale E Sebelisoang Mahaeng Maseru.
	1	2	3	4	5	Total
Strongly agreeEe, Ee	Agree/Ee	NeutralHa ke nke lehlakore	DisagreeChee	Strongly disagreeChee, Chee	50(100%)
People in this community engage in indigenous systems and practices of SWM.Baahi ba mona ba sebelisa mekhoa ea khale ea ho laola lithole			X			
People in this community engage in indigenous systems and practices of SWM because it is a good thing to do.Baahi ba mona ba sebelisa mekhoa ea khale ho laola lithole hobane ke ntho e nepahetseng.					X	
The indigenous systems and practices of SWM in this community is based on the principle of WM hierarchy.Mokhoa oa khale oa ho laola lithole o etsoa ho ipapisitsoe le taolo ea lithole ea sejoale-joale.		X				
The indigenous systems and practices of SWM in this community is a historical issue.Mokhoa oa khale oa ho laola lithole ho baahi ba mona ke nalane.	X

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
