# Peer review of "Methodology to Investigate Indigenous Solid Waste Systems and Practices in the Rural Areas Surrounding Maseru (Kingdom of Lesotho)"

_ijerph, 2021, doi:10.3390/ijerph18105355_

Round 1

Reviewer 1 Report

The paper (Study Protocol) presents an interesting and recently challenging topic in the field of waste management in Lesotho. However, some text parts and results presentation need (major or minor) improvements. Therefore, a major revision of the paper is suggested before the paper can be published in the IJoERaPH journal. Particular comments and drawbacks that can lead to increasing quality of the manuscript are listed below.

Major comments:

  • The title is too long and thus almost unreadable. It has to be shortened and has to involve only crucial information.
  • For the abstract, I suggest you summarize what you have done at first and then you write what is the future research plan. Also, abstract has to be able to stand alone, however, reading and understanding of this abstract is very difficult. Similarly, conclusions.
  • Major revision of the language (proofreading) is strongly suggested.
  • Continuity of the text must be better (when reading I don’t understand many important parts and terms).
  • There is an enormous existing number of literature sources that deal with similar or different research. The literature review involved in this manuscript lacks some important studies and research directions (preferably recently published) that can be inspiring for the research described. As an inspiration, the following research directions and their review can strengthen the research plan and findings, e.g.:

- Circular economy perspective (https://doi.org/10.1016/j.scitotenv.2019.134652) and its connection to the waste network design planning (https://doi.org/10.1007/s10100-019-00626-z),

- A survey of strategic and tactical decisions for the WM (https://doi.org/10.1016/j.cor.2013.10.006),

- Waste infrastructure planning and location of facilities and bins (https://doi.org/10.1016/j.cor.2014.05.003)

- Sustainable waste processing (https://doi.org/10.1016/j.energy.2020.118257),

  • Scheme 2 is not readable. Regarding its caption – is not there a mistake related to the section numbering?
  • Tables formatting doesn’t follow journal rules.
  • There are unacceptable mistakes regarding the work with references and citations (especially, the list of references and its formatting are not unified).

Author Response

Good morning

Please find my response as suggested

Regards

MF Senekane

Reviewer 2 Report

Find my comments below:

ABSTRACT

rural areas under study will be 693 from total population of 6917 – EDIT the rural areas are not that much, it is persons interviewed…

STATKON will analyse data – will be used – it is not a person but a program

the system and practices – must read: the systems and practices

gap in SWM, which, favours the existence - must read gap in SWM, which favours the existence

introduction

…such as waste collection services and sanitation facilities and – should read ….. such as lack of waste collection services and sanitation facilities

P2

… properly [2]. end with full stop

background on - background to

rural communities of Maseru – explain this – Maseru is a city? Maybe communities surrounding Maseru?

indigenous practices of WM in developing and developed – this was not done.

twenty rural areas of Maseru – must read - twenty rural areas surrounding Maseru (see abstract mentioning - 693 study areas

using the note pad, camera and, voice recorder – mention that these are instruments

exception to the existing theoretical framework – what is it currently?

various rural communities of Maseru – must read: 20 rural communities of Maseru

p3

when discharging their various waste activities – replace ‘discharging’ with a more suitable word like executing / performing

It is suggested that a better copy of the map be used. The map must indicate all the selected communities

P4

will randomly be selected from – must read: will be randomly selected from

p5

make sure the table aligns with the requirements of the journal.

Edit: analysed separately. will be analysed qualitatively and quantitively

Edit to complete the sentence adding an action: maybe will not be included?

Community members (CMs) under the age of 21 and over the age of 65, CMs who are blind are not able to use their vision in any meaningful way, visual impairment affects their ability to read printed material, thus reducing their ability to gain information and thereby making him more dependent on others.

P6

respond to same questions – must read: respond to the same questions

Pilot study will take – must read: The pilot study will take

and observe their daily activities those – clarify and edit: I read that they are going to sit around from 8-16:00 at a household to observe their daily activities.

Pri-mary school is chosen must read: A pri-mary school is chosen

conduct 6 to7 interviews – add a space before 7

households will be divided into groups – clarify

recording devise – must read recording device

of 17 to 19 out of 693 questionnaires will be administered every day on – above it was 6-7 please clarify.

Timeframe for this – must read: The timeframe for this

Two – must read two

Let table start on new page

The total is 1732 where does this number comes from?

Why is ‘Example of Quantitative questions’ used? Mention of it is for the pilot study.

P7

refers to a study of material data has been organized - must read: … refers to a study of material data which has been organized

p8

into significant and understandable using – must read: into significant and understandable data using (edit - word missing)

Data will be analysed in terms of …….. literature and new findings of the study – will you analyse the literature? Will you analyse findings? – it does not sound right.

percentage will be used to communicate how a group of participants within figures relates to a larger group of participants. Edit: this does not make sense.

and explain it afar data points and analysis - Edit: this does not make sense.

STATKON (University of Johannesburg statistician - is this a person or a programme? A statistician refers to a person.

Reduce the overuse of “The researcher will….”

…findings of the previous studies - which studies, or must it read: of previous studies …?

training of research field workers, the training will focus on to count a behaviour or responses. - remove spaces

The questionnaire was taken to STATKON to evaluate it for reliability and validity. – delete it is a repeat.

Permission will also be obtained from….. remove or produce the evidence / change to was obtained

….participants will be informed of the purpose, nature, how data will be collected and the extent of the research. The questionnaire will have a tick …etc this must not be in the future tense.

P9

Coerced – edit word unknown

……….concerned – edit sentence: maybe concerns?

Participants – must read: participants

This sentence is unnecessary – it sounds as if there are some unlawful activities bit only not with the people: The study is not intended to have influence on anybody by way of being engaged in any unlawful activity that is related to the study

Repeat – edit article to prevent this: Participants have a right to decide whether to participate in the study, this means that no one will be forced to participate.

Repeat: All participants meeting the selection criteria will be given a chance to participate in the study.

Trustworthy must read: trust

indigenous system – must read: indigenous systems

remove: The results will then be considered for publication in an accredited journal.

Presentations from the study will be made both nationally and internationally – I was under the impression that the research was done? Please clarify.

It is unclear why the is included:

Each research question assessed and critically analysed can help identify the specific weaknesses and strengths in indigenous system and practices of solid waste management in the rural communities of Maseru. Where weaknesses are identified, mitigation measures can be evaluated and implemented to rectify the negative aspects and improve the system and practices.

Brewery – should read: brewery

Discussion – no background is given where these stats are coming from….

P10

This table is confusing as 100% is not calculated for each category. Additionally, tannery, mentioned a s a large generator of waste is not mentioned in the table. Is blasting sand part of textiles? As is pumice stones? And with such a high %?

Below the table a discussion is on the 2012 waste in Maseru with no reference to the background to this or by whom it was done. Why is it included as the study is about the rural MASERU.

…vegetables and, certain items. – Edit – it is uncommon to have it at the end of the sentence.

Thotobolo should be demarcated from the…. – EDIT: I demarcated the correct word?

P11

high (55-80%) quantities of waste …. should read high quantities of waste (55-80%) are

other clean papers are used for packaging fruits, vegetables, and certain food items - repeat and must be removed.

0.09kg/ day while Seychelles generates more waste than Ghana to a volume of 2.98kg/day. – is this per day per person?

23.21; 18.35 and 17, 45 million tons – tons is not comparable with the previous statistics.

This entire section is a literature review and not a discussion as indicated in the heading. This section should be moved to above the methodology discussed earlier. All repeats indicated in yellow in pas 2 on p 11 must be removed:

The total population of the rural areas of Maseru under study is 6,917. The sample size for the rural areas under study will be 693 A pilot study to test the research tools will take place at Ha Teko and this will include 50 participants selected randomly. Participants who will par-ticipate in the pilot study will not form part of the main study. Primary data will be collected through observations and the researcher. Questionnaires written inSesotho and English will be hand-delivered and will be collected after two days. Data will be analysed quantitatively with the assistance of STATKON at the University of Johan-nesburg by using International Business Management Statistical Package for Social Sci-ence version 25.0. Descriptive method will be used to interpret the results. For validity, the interview questions will be set towards answering the research questions in this pro-posal. For reliability, the researcher will design open and close-ended questions. The ques-tions are set in such a manner that they produce stable and consistent results. Permission to conduct the study will be obtained from the Departmental Research committee, Higher Degree Committee and Research Ethics Committee of the University of Johannesburg, the chiefs and, the Ministry of Environment in Lesotho. This study will make recommenda-tions to the Ministry of Environment to work with the communities to plan and imple-ment waste management and development projects. The findings will be published in an accredited journal and will be presented at national and international level in conferences, seminars and special events organized in the study area.

define research paradigm as a guideline used to construct and implement research systems. According to [19], research paradigm uses ontology, epistemology, and methodology to define the nature of a phe-nomenon. An indigenous research paradigm believes that knowledge is relational; there-fore, the researcher should consider sharing knowledge that will emanate from the find-ing of the study with participants, government of Lesotho and the University of Johan-nesburg. The Philosophy that underpins this paradigm is the Critical Theory and in-formed knowledge on indigenous systems [20].

It is my opinion that this section on IRP (2nd paragraph) is more suitable to be used in this article to explain what was done that the entire long section above explaining the methodology etc. of the study.

Final observations

It is not clear how the title is manifested in the script. There is no evidence of a ‘CRITICAL ANALYSIS’. Your opinion on the matter of waste around Maseru was not critically analysed.

An extensive description was made on how the study was done, but no results were given at all. A one-sentence reference is made to how waste is handled in the rural areas. The outcome of the focus groups and the questionnaires were not discussed.

The impact on the environment and human wellbeing was not discussed but only a general reference was made towards it. Identifying appropriate control measures were not suggested.

Was the purpose of the article perhaps to discuss a methodology to investigate indigenous systems?

The impression is that it is a section of a Master’s study.

It is my opinion that this article cannot be published as is, unless it is adapted to be a critical analysis, or that the title changed to “Methodology to investigate rural waste practices in the rural areas surrounding Maseru”. If this option is chosen, the article must be adapted to reflect only this, and literature added to support the methodology.

Author Response

Good morning

Please find my response as suggested.

Regards

MF Senekane

Round 2

Reviewer 1 Report

Thank you for submitting the revised paper. I think that the major revision led to increasing quality of the manuscript. At the moment, I think that only several minor revisions need to be addressed before the paper can be accepted and published in this journal.

Minor comments:
- authors should improve their work and using of abbreviations (e.g., once WM and SWM is defined, don't use "waste managegement" and "solid waste management" respectively);

- authors should also improve their referencing style (the references are not unified: e.g., 1. somwehere is the DOI number, somwehere is the DOI web link, somwehere the DOI number completelly misses, 2. somwehere you refer to both first name and surname, somwhere you wrongly use firstname instead of surname, e.g. in [27]). Please always strictly follow referencing style of the journal, where you want to publish!

- you should also improve the work with sections. For example, if you dont have any 3.2 subsection, then you should not have any 3.1 subsection (it should perhaps only be section 3 instead).

Author Response

Good morning

Please find my response for round 2 comments

Regards

MF Senekane

Reviewer 2 Report

The article makes more sense now with the explanation provided.

Lines 110 to 131: it should be clearly indicated that this is the hazardous components in general waste. At the moment it is not presented in context – there should be very little of this present in the domestic waste and, it must be stated where it will be found within the general waste (like empty discarded bottles which contains residues of cleaning agents). Reference should be made that general industrial waste might also contain hazardous components. All above should have references.

Line 87 – density of waste is not used correctly here in household context and should be removed.

98 - Literature has shown that low-income communities generate waste that is high in

moisture content. – add a references here.

178 - Thotobolo should not be considered or classified… this is not clear – should it or should it not? Edit the sentence to clarify it.

182 should be referenced

260 “Comprehensive studies – I cannot find the close of this quotation.

273 –174 the use of inverted commas is not clear – only the first phrase is used correctly then one open and one un-closed ones follows.

299-304 such long quotations should be avoided – re-write into own language or discard.

306 and 369 unless specified for this paper, the figures are usually named below the figure not above. Are the figures original? If not, it needs a reference. Unless I missed it, I do not see any reference of the article in the text.

329-335 such long quotations should be avoided – re-write into own language or discard

343-358 such long quotations should be avoided – re-write into own language or discard

431-435 – the reference to questionnaires is repeated below and should be deleted where the design is discussed.

Maps should have the annotation below the map.

606-607 edit to make sense: …..or the person who in-charge of cleaning

687 Table does not have a name on top and green script is found in the table.

888 – evidence of a link should be referenced…

Reference list – you must still reference the date accessed

The reference list should be checked to ensure all in-text references are in the list and vice versa.

It is recommended that the supervisor agree with this study by providing proof of such approval.

Author Response

Good morning

Please find attached my response to comments for round 2 of the article.

Regards

MF Senekane
